# Comparative study of neighboring Holm oak and olive trees-belowground microbial communities subjected to different soil management

Antonio J. Fernández-González[1]*, Nuria M. Wentzien[1], Pablo J. Villadas[1],
Antonio Valverde-Corredor[2], Ana V. Lasa[1], Carmen Gómez-Lama Cabanás[2],
Jesús Mercado-Blanco[2], Manuel Fernández-López[1]*

1 Departamento de Microbiología del Suelo y Sistemas Simbióticos, Estación Experimental del Zaidín, Consejo Superior de Investigaciones Científicas (CSIC), Granada, Spain, 2 Departamento de Protección de Cultivos, Instituto de Agricultura Sostenible, CSIC, Córdoba, Spain

* manuel.fernandez@eez.csic.es (MFL); antonio.fernandez@eez.csic.es (AJF)

## Abstract

It is well-known that different plant species, and even plant varieties, promote different assemblages of the microbial communities associated with them. Here, we investigate how microbial communities (bacteria and fungi) undergo changes within the influence of woody plants (two olive cultivars, one tolerant and another susceptible to the soilborne fungal pathogen *Verticillium dahliae*, plus wild Holm oak) grown in the same soil but with different management (agricultural *versus* native). By the use of metabarcoding sequencing we determined that the native Holm oak trees rhizosphere bacterial communities were different from its bulk soil, with differences in some genera like *Gp4*, *Gp6* and *Solirubrobacter*. Moreover, the agricultural management used in the olive orchard led to belowground microbiota differences with respect to the natural conditions both in bulk soils and rhizospheres. Indeed, *Gemmatimonas* and *Fusarium* were more abundant in olive orchard soils. However, agricultural management removed the differences in the microbial communities between the two olive cultivars, and these differences were minor respect to the olive bulk soil. According to our results, and at least under the agronomical conditions here examined, the composition and structure of the rhizospheric microbial communities do not seem to play a major role in olive tolerance to *V. dahliae*.

## Introduction

Olive (*Olea europaea* L. subsp. *europaea* var. *europaea*) cultivation is part of the history and culture of the Mediterranean Basin countries where this tree constitutes an outstanding agro-ecosystem [1]. In Spain, the world's largest olive oil and table olive producer, this woody crop has indisputable social, economic and agro-ecological relevance. In fact, this country accounts for almost 25% of the total olive trees and produces more than 37% of the world's olive oil

**Data Availability Statement:** The datasets generated and analyzed during the current study are available in the NCBI Sequence Read Archive (SRA) under the BioProject number PRJNA510560

**Funding:** This research was funded by grants AGL2016-75729-C2-1-R from the Spanish Ministerio de Economía, Industria y Competitividad and PID2019-106283RB-I00 from Spanish Ministerio de Ciencia e Innovación/Agencia Estatal de Investigación, and co-financed by the European Regional Development Fund (ERDF).

**Competing interests:** The authors have declared that no competing interest exist.

**Abbreviations:** Fra, cultivar Frantoio; HOB, holm oak bulk soils; HOR, holm oak rhizosphere soils; OLB, olive orchard bulk soils; OLR, olive trees rhizosphere soils; PGPM, plant growth promoting microorganisms; Pic, cultivar Picual; VWO, Verticillium Wilt of Olive.

(http://www.internationaloliveoil.org/). In Andalusia (southern Spain), with 60% of the Spanish and 30% of the European olive cultivation area (https://www.mapa.gob.es/es/estadistica/temas/estadisticas-agrarias/agricultura/), the historical impact of a steady expansion of this woody crop on indigenous Mediterranean forests has been significant. Indeed, areas originally covered with oak forests in this region have sharply decreased in the last centuries due to, among other major reasons, the continuous increase of land devoted to olive cultivation, and also because of the peculiar Holm oak (*Quercus ilex* L. subsp. *ballota*) agroecosystem implemented in some regions of the Iberian Peninsula: the so-called in Spanish 'dehesas' [2].

In most of the olive orchards excessive plow, clearing and phyto-sanitary products are often used. These agricultural practices lead to a decrease in biodiversity [3,4] which is also reflected in the microbial communities associated with these woody plants [4,5]. For example, the use of pesticides affects soil microbial and functional diversity, as they can promote or decrease the growth of certain microorganisms and alter metabolic pathways [6,7]. In addition, fertilizers can disturb microbial communities as they change the nutrient content of the soil [8]. Furthermore, the type of management has an important influence in different ways: tillage can produce desiccation, soil compaction or mechanical structure disruption among others. They can also affect the availability of plant residues in the soil and thus the soil organic matter and distribution of nutrients [9].

All these practices influence microbial communities and can deeply affect tree health and development [10], contributing to the expansion and severity of different abiotic (e.g. soil erosion, drought, salinity) and biotic (e.g. Verticillium wilt of olive [VWO], the decline of Holm oak, *Fusarium*-caused infections and others pest expansion) stresses very difficult to handle, particularly under the climatic conditions predominant in Andalusia (irregular and scarce rainfall with prolonged drought periods) [11]. Nowadays, VWO is considered one of the most relevant disease in many regions were olive is cultivated. It is a vascular disease very difficult to control, caused by the soil-borne fungus *Verticillium dahliae* Kleb. Because of the relevance of olive cultivation in the Mediterranean Basin (and the expansion of this tree crop to other regions worldwide), studies on the epidemiology and management of VWO are of utmost importance [12]. In many cases, olive growers try to overcome loses produced by the disease by removing affected or dead trees of susceptible cultivars (i.e. Picual) and replacing them by new plants of varieties displaying tolerance to the pathogen (i.e. Frantoio) [11]. Our knowledge on the tolerance/resistance mechanisms of 'Frantoio' trees to *V. dahliae* has increased from transcriptomics, biochemical and histological approaches [13–16]. However, these studies only refer to the role of the host genetics. In contrast, the potential involvement of the olive root associated microbiota in VWO tolerance has not yet been sufficiently evaluated, and only very recently under conditions in which the pathogen is artificially inoculated in nursery-produced olive plants [17].

It is well-known that the soil is one of the Earth's environments with the highest level of microbial diversity [18–21], and that the rhizosphere is the soil hotspot in which most of the plant-microbe interactions take place [22]. These interactions could result in neutral or deleterious effects for the plant, although a number of plant-associated microorganisms provide beneficial effects (e.g. increasing yield or reducing abiotic and biotic stress) to the host, so they are generally known as Plant Growth Promoting Microorganisms (PGPM). While plant growth promotion is more related with microbial activity in the rhizosphere, bulk soil microbial communities are also relevant in terms of soil fertility. In general, these microorganisms participate in litter decomposition, nutrient cycling degradation of pesticides and pollutants, and organic matter dynamics, among others [5,6,8,22]. Because of that, studies on soil microbiota are increasing with the aim to identify microorganisms with biotechnological interest [23,24].

In contrast to the Holm oak-associated microbiome, well-defined in different environmental conditions by high-throughput sequencing technologies [25–29], our knowledge on olive-associated microbiota is still very scarce and fragmentary [30–37]. In fact, only few studies are available on bacterial or fungal communities associated with specific olive organs or the olive rhizosphere, and most of them are based on non-high-throughput approaches. For example, Aranda *et al.* [38] studied bacterial communities associated with wild olive (*Olea europaea* L. subsp. *europaea* var. *sylvestris*) roots, but using fluorescent terminal restriction fragment length polymorphism (T-RFLP) and bacteria isolation in culturing media. In another study, Martins *et al.* [34] carried out a screening of fungal communities by a culture-dependent method in olive trees of cultivar Cobrançosa. However, a new study has been recently performed [39] describing the belowground microbial (bacteria and fungi) communities of 36 olive cultivars but, it was only focused on an experimental field without soil management comparison.

On the one hand, different studies have shown how microbial communities are influenced by land use (agricultural with or without tillage, grassland, etc.) [40,41] along time [42], soil characteristics [43], or even to the geographical distance at European scale [44]. These effects have been mainly described for herbs and grasses and are less documented for tree species grown in the same soil [45]. Moreover, there is total lack of information on this regard for varieties or cultivars from the same tree crop differing in tolerance to plant pathogens when cultivated under field conditions. On the other hand, wild conditions are not normally considered and if so, studies are not performed at the rhizosphere level or they only implemented culture-dependent techniques [46–52]. Since the rhizosphere is a hotspot for microbial diversity and for PGPM, it is worth exploring microbial communities residing in this particular niche (as well as in the bulk soil) in both olive orchards and their neighboring wild woods. Important questions to be addressed would be whether microbial communities from the rhizosphere of wild (e.g. oaks without agronomical practices) woody plants growing in spots adjacent to cultivated (e.g. olive orchards subjected to usual agronomical management) trees could be a source of relevant microorganisms for the latter ones. Moreover, this inquiry must also be carried out for the bulk soil in order to find links between soil management and microbial communities that could help to improve olive growing conditions. Finally, in addition to soil management, the analysis of the rhizosphere effect may also serve to uncover key microorganisms relevant for the health status of these woody plants.

Taking into account the above information, we aimed to study whether rhizosphere and bulk soil microbial (bacteria and fungi) communities associated to perennial, long-living woody plants undergo changes when growing in neighboring plots submitted to either none (Holm oak) or agricultural (olive) soil management practices using deep-sequencing of the *16S rRNA* gene and the ITS2 region amplicons. We also want to test the hypothesis that the olive-associated soil microbiota under agricultural practices is differently shaped depending on the VWO susceptibility level of the olive cultivars (Frantoio, VWO tolerant; Picual, VWO susceptible) present in a commercial orchard established in land gained to a Holm oak forest, and in which the disease has been previously present.

## Materials and methods

### Soil sample collection

Soil samples in the olive orchard were collected when olive trees were in full bloom. The surveyed olive orchard was established decades ago in land gained to a wild Holm oak forest over a Cambisol soil, and it is currently surrounded by Holm oaks and accompanying native herbs and shrub vegetation. The sampled site is located in the municipality of Mancha Real, Jaén province, Spain (37˚53'16.8" N; 3˚38'06.1" W; 480 meters above sea level) and the sampling

was carried out in the spring of 2018 (S1 Fig in S1 File). Permission to get access to the orchard was granted by the owner of the farm, who was informed of the sampling activities in advance. Additional permits from local, regional or national authorities were not needed since no attempt to collect genetic resources was intended. The olive orchard was under traditional management at least for 15 years, including mechanical plow and herbicides and pesticides treatments according to the common extension worker recommendations implemented in the region. In contrast, Holm oaks are not subjected to any agronomical practice. The orchard was originally established with 'Picual' plants while 'Frantoio' trees were planted to substitute dead 'Picual' trees due to VWO attacks. For each olive cultivar (Picual and Frantoio respectively) and Holm oak trees, five replicates were collected, that is, five trees were selected for sampling. Moreover, the sampling of each replicate was made in a way to increase the representativeness of the microbial communities. With that intention, each tree was sampled in two opposing sides (N = North side, S = South side), avoiding the zone directly influenced by the drippers of the irrigation system present in the orchard. The upper soil layer (first 5 cm) was removed and rhizosphere soil samples were collected (5 to 20-cm depth) following the main roots of each plant until finding young, cork-free roots. At each sampling side (N and S), two digs (N1, N2 and S1, S2) were performed to find and collect the soil firmly adhered to active roots. The soil of each dig from the same side of the tree was mixed (20 g) and stored at 4 ˚C, so for each tree (that is, each replicate), two DNA extractions were carried out within 24 hours after sampling (see section 2.2). Finally, the two DNA extractions of each tree (N and S) were equitably mixed (S2 Fig in S1 File). The 15 DNA samples (5 Picual, 5 Frantoio and 5 Holm oak) were labeled before sending to massively parallel sequencing. Samples of bulk soil were also collected between Holm oak trees with five replicates and at the middle point between neighboring sampled 'Picual' and 'Frantoio' trees (500 g each of the five replicates). Therefore, 5 DNA samples from each of the 5 treatments (n = 25) ['Picual' (Pic), 'Frantoio' (Fra), and Holm oak (HOR) rhizospheres, olive bulk soil (OLB) and Holm oak woods bulk soil (HOB)] were sequenced. Furthermore, bulk soil samples were also analyzed at the Agri-Food Laboratory of the Andalusian Regional Government (Córdoba, Spain) to determine physicochemical properties using standardized procedures implemented in this service (S1 Fig in S1 File).

## DNA extraction and Illumina sequencing

The soil DNA from each individual sample was obtained using the Power Soil DNA Isolation kit (MoBio, Laboratories Inc., CA), following the manufacturer's recommendations within 24 hours of samples collection. DNA yield and quality were checked both by electrophoresis in 0.8% (w/v) agarose gels stained with GelRed and visualization under UV light, and by using a Qubit 3.0 fluorometer (Life Technologies, Grand Island, NY). The DNA was sequenced with the Illumina MiSeq platform at the genomics service of the Institute of Parasitology and Biomedicine "López Neyra" (CSIC; Granada, Spain) following the recommended Illumina's protocols. In the first run, a prokaryotic library was constructed amplifying the hyper-variable regions V3-V4 of the *16S rRNA* gene using the primer pair Pro341F (5'-CCTACGGGNBG CASCAG-3') and Pro805R (5'-GACTACNVGGGTATCTAATCC-3') according to Takahashi *et al.* (2014) [53]. These amplicons were tagged to be attached to PNA PCR clamps to reduce plastid and mitochondrial DNA amplification [54]. In the second run, a eukaryotic library was constructed amplifying the ITS2 region using the primer pair ITS4 (5'-TCCTCCGCTTATTGATATGC-3') [55] and fITS7 (5'-GTGARTCATCGAATCTTTG-3') [56]. Both runs were sequenced using a paired-end 2x300-bp (PE 300) strategy. Moreover, a ZymoBIOMICS Microbial Community Standard II (Log Distribution), ZYMO RESEARCH (https://www.zymoresearch.com/collections/zymobiomics-microbial-community-standards/

products/zymobiomics-microbial-community-standard-ii-log-distribution), was added in each run as metabarcoding sequencing control.

## Data quality screening and overlapping

Demultiplexed and Phi-X174-free reads were quality checked with FastQC v.0.11.5 (http://www.bioinformatics.babraham.ac.uk/projects/fastqc/) and end-trimmed with FASTX-Toolkit v.0.014 (http://hannonlab.cshl.edu/fastx_toolkit/index.html). All the 3'-end nucleotides were removed until the first position which reached an average quality value higher than Q25. The paired reads were overlapped with fastq-join v.1.3.1 [57] requesting a minimum overlap of 40 bp and a maximum of 15% of difference in the overlapping region. Both libraries were processed with the same bioinformatics tools but following different pathways detailed below.

## Prokaryotic data processing

Using the software SEED2 v.2.1.05 [58] the prokaryotic sequences were trimmed and clustered. Firstly, by trimming the specific primers; then, by removing sequences with ambiguities and shorter than 400 bp as well as reads with an average read quality lower than Q30. Secondly, chimeric reads were removed by VSEARCH "De Novo" v.2.4.3 [59] implemented in SEED2 and OTUs were clustered with the same tool at 97% similarity. Finally, the OTU table was saved and OTUs accounting for less than 0.005% of the total sequences were removed according to the MOCK community used [ZymoBIOMICS Microbial Community Standard II (Log Distribution), ZYMO RESEARCH] and Bokulich *et al.* [60] for further analyses. The most abundant OTU sequences were retrieved in SEED2 and classified with an 80% bootstrap cutoff to the Ribosomal Database Project (RDP-II) 16S rRNA reference database, training set v.16 MOTHUR-formatted [61], with MOTHUR v.1.39.5 [62]. The OTUs classified as mitochondria, chloroplast and unknown (unclassified at kingdom level) were removed from the OTU table. This classification was considered as the taxonomic information of each OTU.

## Eukaryotic data processing

The eukaryotic library was quality-trimmed in SEED2 by the removal of sequences with ambiguities and an average read quality lower than Q30. The specific primers and those sequences smaller than 100 bp were removed. Subsequently, chimeric sequences were identified and discarded with VSEARCH "De Novo" implemented in SEED2. Then, the good quality sequences were distance-based greedy clustered at 97% similarity with VSEARCH algorithm implemented in MOTHUR. The most abundant OTU sequences were classified using the UNITE v.7.2 dynamic database [63] with MOTHUR following the parameters recommended in the website and used by Findley *et al.* [64]. Finally, only OTUs with more than 0.005% of the sequences, according to the MOCK community used, and assigned to kingdom Fungi were kept for further analyses.

## Construction of core microbiomes

The bacteriome and mycobiome core taxa were generated considering only genera that were present in all the replicates (n = 5 and n = 10 in the case of OLR) of each treatment. Shared genera were present in both compared conditions, and the specific ones were present in its treatment but missing in at least one replicate of the other condition. After construction, core taxa were plotted in Venn diagrams. Core taxa and Venn diagrams were performed in MS Office 2016 Excel tool.

## Statistical analyses

α-diversity indices [Observed and Chao1 richness; Shannon and inverse of Simpson diversity (InvSimpson)] were compared with Kruskal-Wallis test applying Mann-Whitney-Wilcoxon as a *post hoc* test and *p-value*s were FDR corrected by the Benjamini-Hochberg method using the R package *agricolae* [65]. Concerning the β-diversity, a normalization of the filtered OTU sequence counts was performed using the "trimmed means of M" (TMM) method with the BioConductor package *edgeR* [66]. The normalized data were considered to perform Principal Coordinates Analysis (PCoA) on Bray-Curtis, Unweighted and Weighted Unifrac distances to ordinate in two dimensions the variance of β-diversity among all treatments in prokaryotic dataset, and only Bray-Curtis dissimilarities were used in eukaryotic dataset. In order to use Unweighted and Weighted Unifrac distances, a phylogenetic tree was produced using the online tool MAFFT [67] and FastTree [68]. Ordination analyses were performed using the R package *phyloseq* [69]. We analyzed the effects of treatment, soil management (natural or agricultural) and kind of soil (bulk or rhizosphere) factors on community dissimilarities with permutational analysis of variance (PERMANOVA) and permutational analysis of multivariate homogeneity of groups dispersions (BETADISPER) using the functions *adonis* and *betadisper* in the vegan package with 9,999 permutations [70]. When applicable, pairwise differences between groups were assessed with the function *pairwise.adonis* from the package *pairwiseAdonis* [71]. Significantly different and biologically relevant prokaryotic or fungal genera were obtained with the following protocol: we tested for differential genus relative abundance using proportions in non-normalized counts with the STAMP v.2.1.3 software [72], selecting default statistical parameters for multiple groups and Welch's t-test with differences between proportions $\geq 0.5$ or ratio $\geq 2$ for two groups comparisons and considering Benjamini-Hochberg FDR for multiple test correction. Those genera significantly different in the two methods previously described were plotted and manually checked to generate the final selection. Furthermore, Constrained Analysis of Principal Coordinates (CAP or dbRDA) with weighted unifrac for prokaryotic and Bray-Curtis for fungal genera were performed, using the function *ordinate*, to see the effect of the physicochemical soil properties and the microbial profiles. Firstly, with the function *capscale* the independent parameters were selected, those with variance inflation factors (VIF) lower than 10. Secondly, ANOVA tests with the function *anova.cca* were performed to obtain the statistically significant parameters in the resultant distribution. Finally, with the function cor.test, the correlation between the significantly different parameters and the genera with $\geq 0.1\%$ relative abundance where computed. Those significant correlations (*p-value* $< 0.05$) with Spearman *rho* $\geq 0.6$ were considered strong positive correlations and those with Spearman rho $\leq -0.6$ were considered strong negative correlations. Most of the steps performed on R were carried out following the R script publicly donated by Hartman *et al.* 2018 [73].

## Data publicly available

The datasets generated and analyzed during the current study are available in the NCBI Sequence Read Archive (SRA) under the BioProject number PRJNA510560.

## Results

### 16S rRNA and ITS sequencing quality

A total of 1,198,153 raw reads were obtained from the prokaryotic dataset. After overlapping, quality trimming and classification, the number of sequences decreased to 816,766, that is a 68.72% of the raw reads. The maximum number of sequences per sample was 40,762 and the

minimum 14,621 which resulted in 810,623 sequences, 99.25% of the final reads, classified as prokaryotes (Bacteria all of them), at least at domain level, and they were clustered in 1,795 OTUs. In the eukaryotic dataset, the number of raw reads was 1,041,312. Overlapping, quality trimming and classification reduced the number of sequences to 807,929 final reads, 77.59% of the raw reads. Thus, the maxium number of sequences per sample was 43,605 and the minimum 21,316. 98.48% of the total sequences were classified as fungi, accounting for 795,647 sequences. They were clustered in 988 OTUs (S1 Table in S1 File). Finally, the mock community sequenced by triplicate for each library showed the following results: A) In the procaryotic dataset 45 OTUs were initally found and just 5 remained (*Listeria* 91.8%, *Pseudomonas* 6.4%, *Bacillus* 1.5%, *Escherichia* 0.1% and *Salmonella*, wrongly annotated as *Buttiauxella*, 0.1%) after crosswalk removal. *Lactobacillus*, *Enterococcus* and *Staphylococcus* (lower than 0.01% in the reference mock) were not detected. B) In the fungal dataset 49 OTUs were initially found and just 2 remained (*Saccharomyces* 99.89% and *Saccharomyces* 0.02%) after crosswalk removal. *Cryptococcus* (lower than 0.001% in the reference mock) was not detected.

## Assessing the bacterial diversity in bulk soils and rhizospheres of olive and Hokm oak

Bacterial α-diversity, after rarefying at 14,621 sequences, was significantly different among groups (*p-value* < 0.05 in Shannon and InvSimpson indices), in contrast to richness (*p-value* = 0.37). More specifically, diversity was significantly higher for Holm oak rhizosphere soil (HOR) (Fig 1A). Good's coverage was not different among samples and always greater than 98%. Moreover, after integrating the information on changes in structure and composition of the communities among different treatments (β-diversity), we also found significant

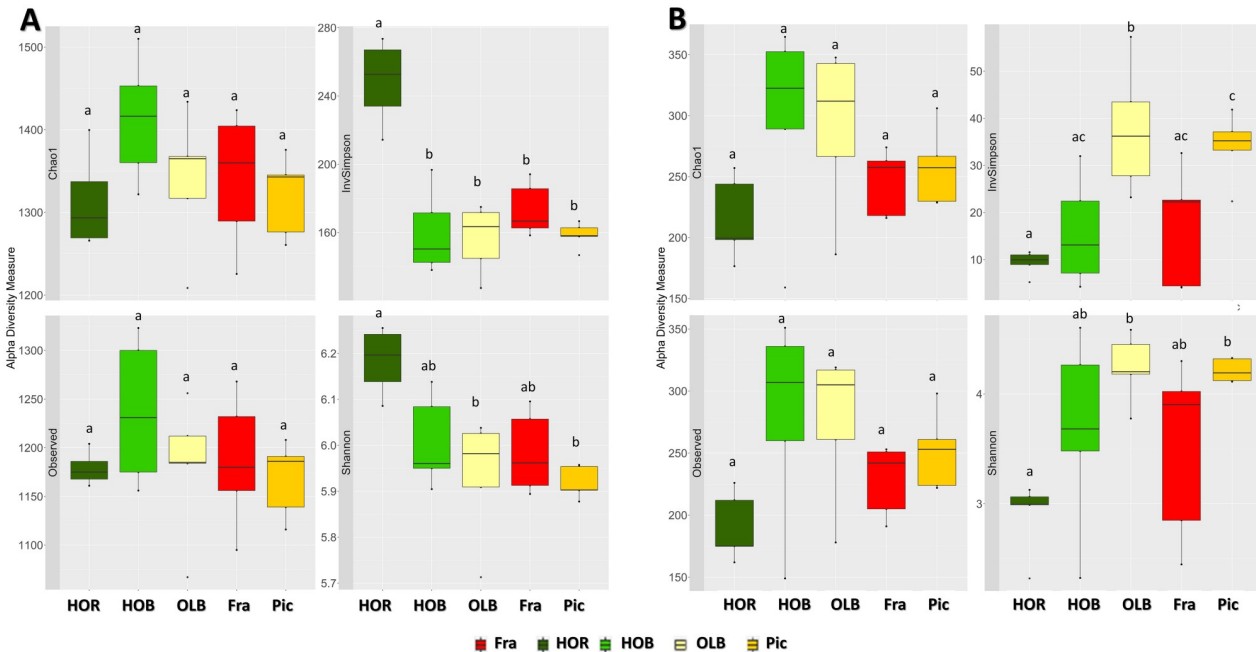

**Fig 1. Indices of microbial diversity.** (A) Bacterial and (B) Fungal α-diversity indices of each treatment. For both panels, it is shown five summary statistics (the median, two hinges and two whiskers) and outlying points. All groups have a sample size of five, except for HOR which has four samples. Pairwise comparisons with statistically significant differences (*p value* < 0.05 with a Mann-Whitney-Wilcoxon test) are indicated with letters. HOR: Holm oak rhizosphere, HOB: Holm oak bulk soil, OLB: olive bulk soil, Fra: 'Frantoio' rhizosphere, Pic: 'Picual' rhizosphere, InvSimpson: inverse of Simpson index.

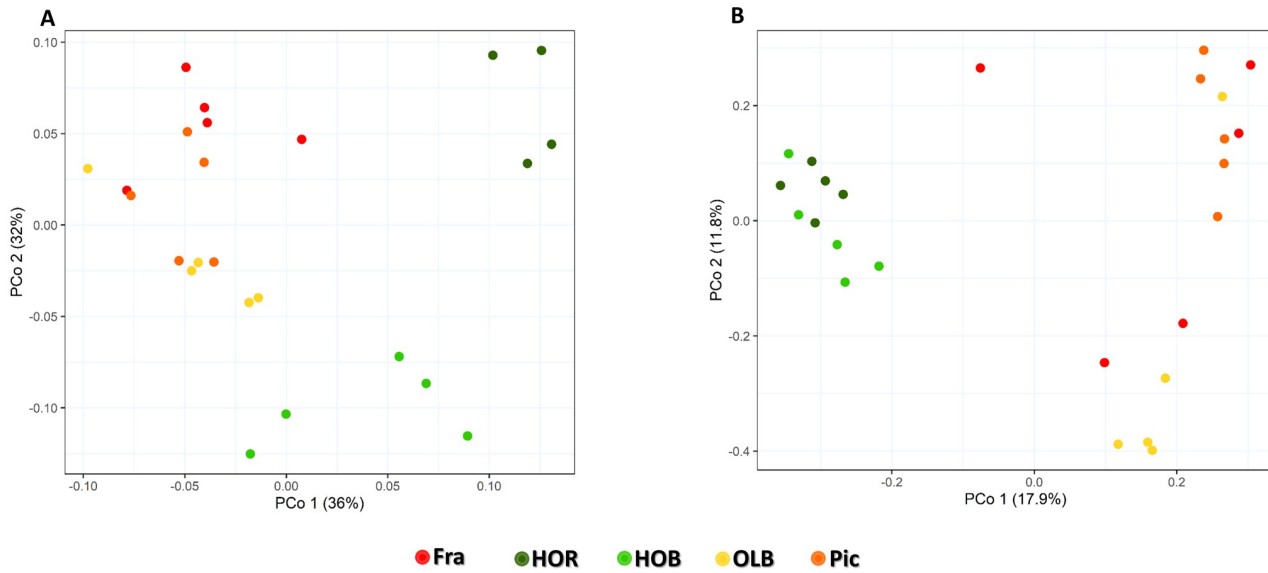

**Fig 2. Principal coordinates analyses of the microbial communities.** (A) PCoA of bacterial communities by treatment on Weighted Unifrac distances. (B) PCoA of fungal communities by treatment on Bray-Curtis dissimilarities. HOR: Holm oak rhizosphere, HOB: Holm oak bulk soil, OLB: olive bulk soil, Fra: 'Frantoio' rhizosphere, Pic: 'Picual' rhizosphere.

differences ($p$-value = 0.001 and $R^2$ = 0.652 with a PERMANOVA test). In order to elucidate the relationship among groups, a BETADISPER test was carried out and variance differences were not statistically significant ($p$-value = 0.965). Graphics were produced using several methods. Among them, PCoA with Weighted Unifrac distances yielded a good visual representation and explained 68% of the variance (Fig 2A). Furthermore, we conducted a pairwise comparison of PERMANOVA test, as a *post-hoc* test, using the same dissimilarity metrics. This test provided significant differences among groups as expected from the PCoA plot. Olive orchard bulk (OLB) and rhizosphere (Fra and Pic) soils were grouped together with no statistically significant differences among their communities, as shown in Fig 2A. Holm oak bulk (HOB) and rhizosphere (HOR) soils seemed to be different in the plot, which was confirmed with the *post-hoc* test. Furthermore, their communities were different from those originating from olive orchards soils. The above mentioned α- and β-diversity analyses were carried out without Holm oak rhizosphere sample 2, as it was identified as an outlier with an abnormally low α-diversity value.

## Estimating the fungal diversity in bulk soils and rhizospheres of olive and Holm oak

To perform the fungal α-diversity analyses, the number of sequences was rarefied to 21,316. Significant differences were found for Shannon and InvSimpson indices (Fig 1B). Pairwise comparison by Kruskal-Wallis *post hoc* test showed that the Shannon index was significantly lower in HOR in comparison to OLB and 'Picual' rhizosphere. Inverse of Simpson index showed the same behaviour. Moreover, it was also significantly higher in OLB in comparison to HOB and 'Frantoio' rhizosphere (Fig 1B). Good's coverage was not different among samples and always greater than 99%. β-diversity was also significantly different among groups according to the PERMANOVA test ($p$-value = 0.001 and $R2$ = 0.358), and BETADISPER was not significantly different ($p$-value = 0.074). The results were plotted with PCoA based on Bray-

Curtis dissimilarities and the two first axes explained 29.7% of the variance (Fig 2B). Fungal communities showed a similar behaviour to that of bacteria with regard to β-diversity in olive soils, but different in Holm oak soils. HOB and HOR communities seemed to be similar between them and different when compared to olive soils and, as observed for bacterial community comparisons, no significant differences were scored between olive soils. Pairwise comparison of PERMANOVA test using Bray-Curtis dissimilaries confirmed these results, except for Holm oak soils comparison, which was significant (*p-value* = 0.016).

## Unearthing differences in the composition of bacterial communities

The taxonomic profile for all groups at the phylum level (Fig 3A) showed *Acidobacteria*, *Actinobacteria*, *Proteobacteria*, *Candidate division WPS-1*, *Verrucomicrobia*, *Bacteroidetes* and *Gemmatimonadetes* as predominant phyla. Altogether they represent at least 80% of the total abundance. In agreement with β-diversity analysis, olive bacterial communities were more similar to each other than to Holm oak communities. *Proteobacteria* was considerably more abundant in Holm oak rhizosphere soil, while *Candidate division WPS-1* was lower in this soil compared to other soil samples. *Gemmatimonadetes* was less represented in Holm oak soils than in olive soils. In line with these results, and at the genus level (Fig 3B), the most abundant genera belonged to *Acidobacteria* (*Gp6* and *Gp4*), *Candidate division WPS-1*, *Actinobacteria* (*Rubrobacter* and *Gaiella*), *Gemmatimonadetes* (*Gemmatimonas*) and *Proteobacteria* (*Sphingomonas*). Once again, taxonomic profiles were more similar between olive soils than between HOB and HOR soils. Holm oak bulk soil was more similar to olive soils than to HOR, although *Gp6* was less abundant in HOB than in the other groups and the opposite was observed for *Gp4* and *Rubrobacter*. Moreover, *Gemmatimonas* was less represented in HOR soils than in olive soils. Finally, other genera changed singificantly like *Gp7*, *Microvirga* or *Mycobacterium*. In agreement with β-diversity, *post hoc* test showed no significant differences (Benjamini-Hochberg FDR corrected *p-values* > 0.05) among phyla and genera when comparing Pic and Fra rhizosphere soils. To get a better insight, two-groups tests were performed between groups that showed significant differences according to the β-diversity analysis and multiple groups comparison. Therefore, Fra and Pic were considered as a single community (olive rhizosphere soil, OLR), that is, Fra and Pic samples were mixed together in one group, so genera average relative abundance was calculated with n = 10 samples instead of n = 5. When comparing HOR and OLR soils (Fig 4A), five genera were significantly different in terms of statistics and biological relevance, namely an *incertae sedis* genus belonging to phylum *Candidate division WPS-1*, *Gemmatimonas*, *Rubrobacter*, *Solirubrobacter* and *GP7*. Interestingly, all of them were more abundant in OLR but *Solirubrobacter* that was more abundant in HOR. The same comparison was performed in bulk (HOB and OLB) soils (Fig 4B). In this case, only two genera showed both statistical and biological significant differences: *Gp4* (more abundant in HOB) and *Gemmatimonas* (more abundant in OLB). Finally, we analyzed differences between HOB and HOR soils (Fig 4C). In this case we found three genera with biologically relevant differences and statistically significant: *Gp4* (more abundant in HOB), *Gp6* and *Solirubrobacter* (more abundant in HOR).

## Unveiling differences in the composition of fungal communities

At phylum level, fungal communities were dominated by *Ascomycota* and *Basidiomycota*, ranging from 63.85% relative abundance in Pic to 91.22% in HOR (Fig 3C). The relative abundance of *Ascomycota* and *Basidiomycota* differed considerably in HOR with respect to the other soils, *Basidiomycota* being the only phylum more significantly abundant. At genus level, the high heterogeneity observed, reflecting a high variability among groups, made it difficult to

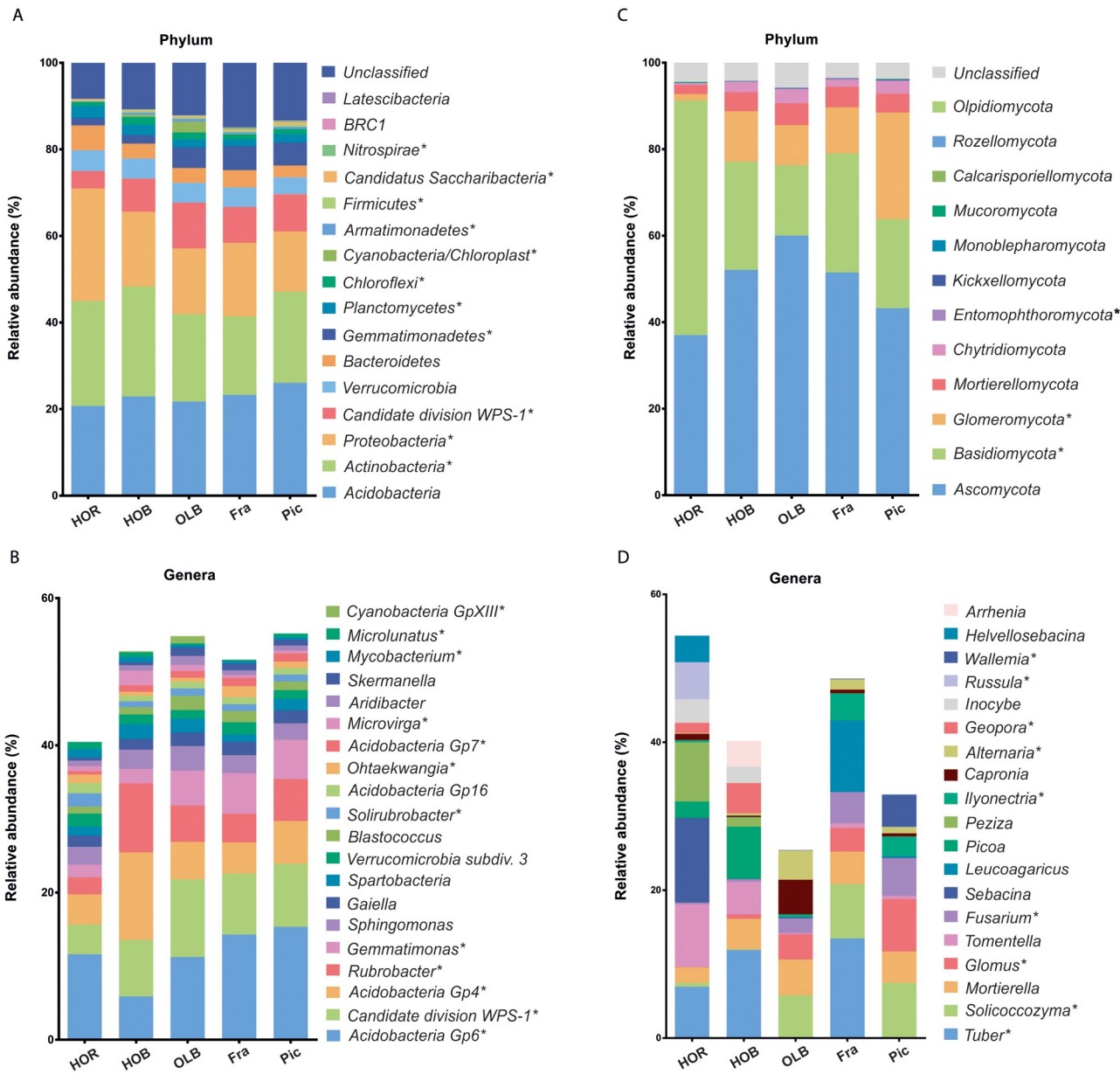

**Fig 3. Microbial taxonomic profiles of soils and rhizospheres.** Bacterial taxonomic profile at phylum (A) and genus level (B); and fungal taxonomic profile at phylum (C) and genus level (D) in the five studied treatments. In panels (B) and (D), only main genera (with a relative abundance ≥ 1%) are shown. Asterisks show statistically significant differences (*p-value* < 0.05) according to ANOVA test and the comparison of 5 replicates in each treatment. HOR: Holm oak, HOB: Holm oak bulk soil, OLB: olive bulk soil, Fra: 'Frantoio' rhizosphere, Pic: 'Picual' rhizosphere.

identify any trend (Fig 3D). A few genera showed large differences between Pic and Fra soils, like *Tuber* or *Glomus*, although they were not confirmed by the *post hoc* test. Therefore, differences were apparently due to very large variability among samples. Because of that, two groups comparison was performed in order to obtain better information regarding statistically significant differences. As for bacterial communities, two groups comparison analysis was performed based on the information obtained with β-diversity and multiple groups comparisons. Thus, rhizosphere (HOR and OLR) soils and bulk (HOB and OLB) soils were compared. When

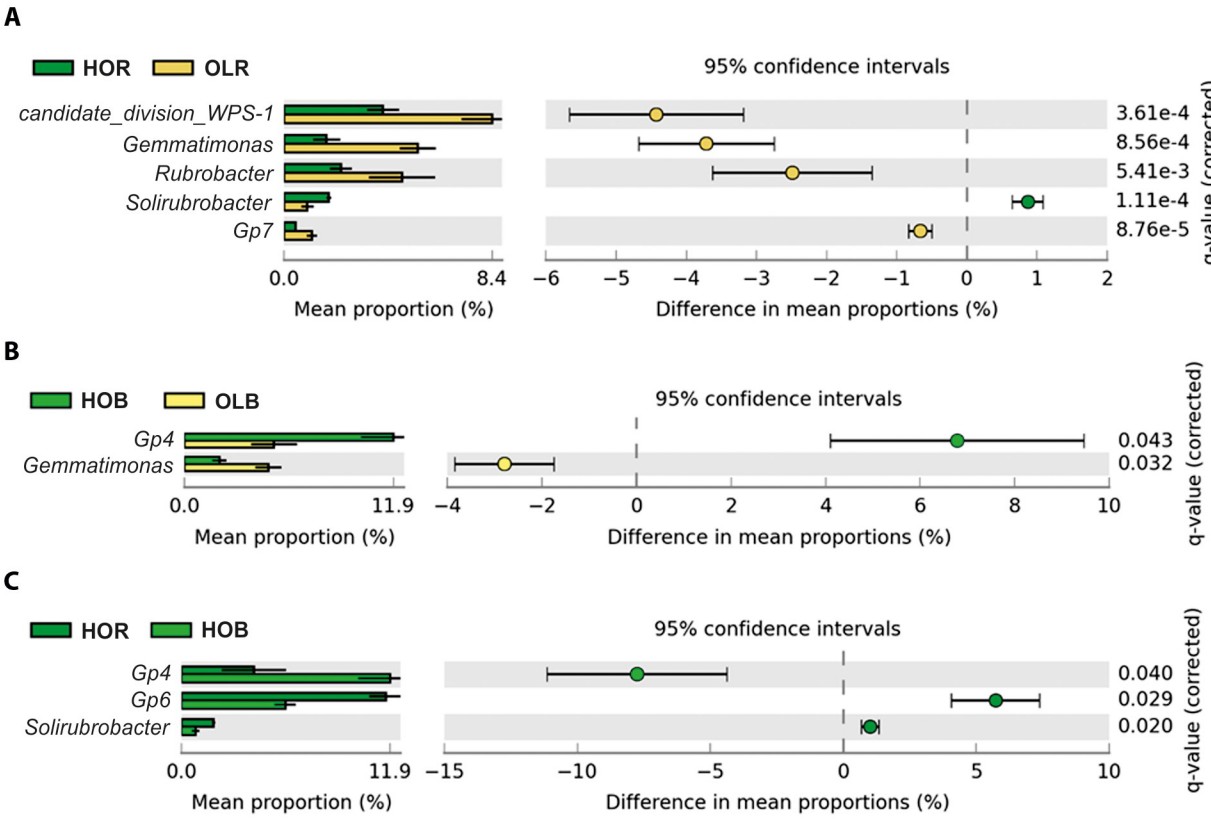

**Fig 4. Genera whose relative abundance differed significantly and biologically among the two-group comparison analyses.** Genera shown had statistically significant differences (*p-value* < 0.05) according to Welch's t-test with differences between proportions ≥ 0.5 or ratio ≥ 2. HOR: Holm oak rhizosphere, OLR: average of 'Frantoio' and 'Picual' rhizospheres, HOB: Holm oak bulk soil, OLB: olive bulk soil.

comparing HOR and OLR rhizosphere soils, four genera showed statistically significant and biologically relevant differences, namely *Solicoccozyma*, *Stachybotrys*, *Subramaniula* and *Fusarium*. All of them were more abundant in OLR soil (Fig 5A). Bulk soil comparison showed only two genera with statistically significant and biologically relevant differences: *Fusarium* and *Coniosporium*, both of them being more abundant in OLB soil (Fig 5B). Finally, this analysis did not show significant differences between HOR and HOB soils with regard to fungal genera, although they displayed very different abundance profiles and differences in the pairwise PERMANOVA test. This result was in accordance with the distribution obtained in the PCoA plot (Fig 2B). Moreover, it has to be mentioned that *Verticillium* was no detected in spite of the deep sequencing analysis.

## Determining core microbiomes from statistically different communities

In the three analyzed cases (HOR vs OLR, HOB vs OLB and HOR vs HOB), the core bacteriome involved a high number of genera with a high relative abundance (S3 Fig in S1 File), being the main genera *Gp6*, *Gp4*, *Rubrobacter* and *Gemmatimonas*. In fact, 99 genera were shared considering rhizosphere soils (57.62% of the sequences from HOR and 65.82% from OLR), 102 genera only for bulk soils (66.78% of the sequences from HOB and 67.03% from OLB) and 103 genera for Holm oak soils (57.67% HOR and 66.49% HOB). Core genera that were specific for each situation were very low in both number and relative abundance.

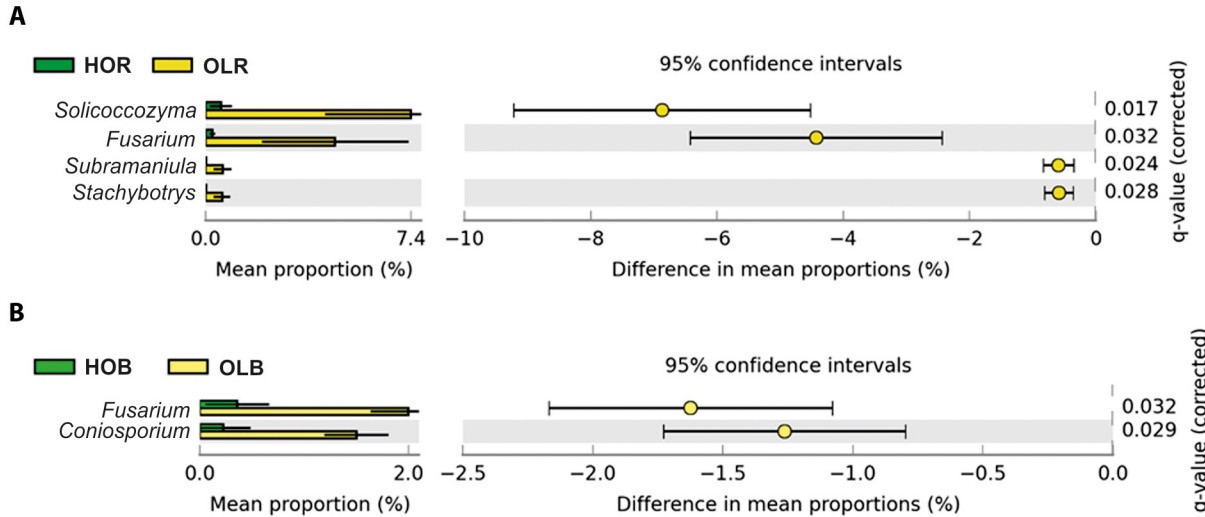

**Fig 5. Genera whose relative abundance differed significantly and biologically among the two-group comparison analyses.** Genera shown had statistically significant differences (*p-value* < 0.05) according to Welch's t-test with differences between proportions ≥ 0.5 or ratio ≥ 2. HOR: Holm oak rhizosphere, OLR: average of 'Frantoio' and 'Picual' rhizospheres, HOB: Holm oak bulk soil, OLB: olive bulk soil.

The core mycobiome was determined in the two statistically different situations. Bulk soils (HOB vs OLB) and rhizosphere soils (BO vs OLR) were taken into account, but not the comparison between treatments from HOR soil (HOR vs HOB) since genera showing significant differences between rhizosphere and bulk soils were not detected. In contrast to bacterial communities, the core fungal genera were low in number and in relative abundance for both situations. The core mycobiome was represented by 13 genera for rhizosphere soils and 18 for bulk soils (S4 Fig in S1 File). In rhizosphere soils, these genera accounted for the 4.94% of the relative abundance in Holm oak and 23.23% in olive, with *Mortierella*, *Solicoccozyma*, *Glomus* and *Fusarium* as the main genera. However, in bulk soils relative abundances were similar: 10.53% for Holm oak and 8.10% for olive, with *Mortierella*, *Capronia*, *Alternaria* and *Glomus* as the main genera. The number and relative abundance of specific core genera of each situation, which had a much bigger relative abundance than in bacteria, is shown in S4 Fig in S1 File.

## Strong influence of soil properties in microbial communities

The influence of soil physicochemical properties (S2 Table in S1 File) on the bacteriome at the genus level was analyzed taking into account all samples together. Physicochemical variables presented as independent were pH, sodium, exchangeable potassium, organic matter and Carbon/Nitrogen ratio. With these variables, the statistical analysis was done on CAP distribution (Fig 6). pH and organic matter differences were statistically significant between the two soil management conditions analysed (native and agricultural) in the distribution of the bacterial communities, CAP1 being the relevant axis. CAP plot gave the best results with weighted unifrac distances, explaining more percentage of variance in the first two axes (35.4%) when comparing to other dissimilarities (Fig 6A). Regarding the correlation between different bacterial genera and significant physicochemical parameters, 20 genera showed a strong and significant correlation (S2 Table in S1 File). Among them, an *incertae sedis* genus belonging to the phylum *Candidate division WPS-1* and *Gemmatimonas* were more abundant in olive soils.

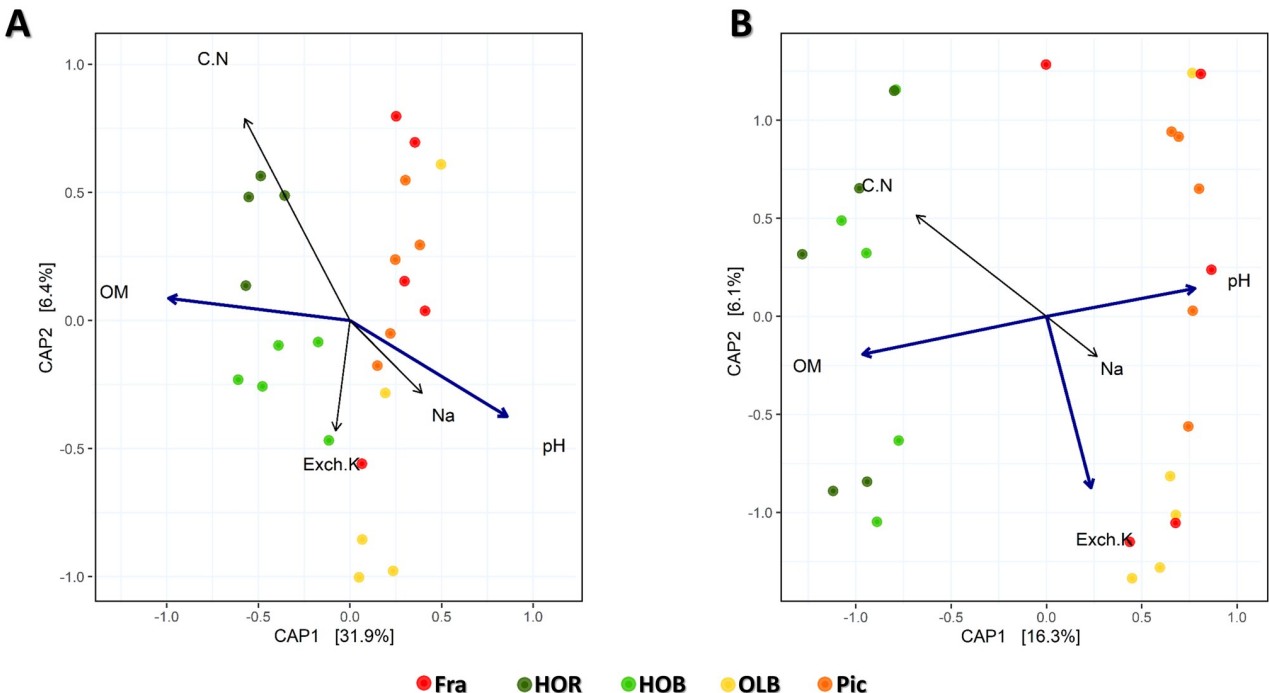

**Fig 6. Constrained Analysis of Principal Coordinates (CAP) of (A) bacterial communities by treatment on Weighted Unifrac distances and (B) fungal communities by treatment on Bray-Curtis dissimilarities; both with all the independent physicochemical parameters.** HOR: Holm oak rhizosphere, HOB: Holm oak bulk soil, OLB: olive bulk soil, Fra: 'Frantoio' rhizosphere, Pic: 'Picual' rhizosphere, OM: organic matter, Exch.K: exchangeable potassium.

Furthermore, a positive correlation with pH was found for *Candidate division WPS-1*, and a positive correlation with pH but negative with organic matter for *Gemmatimonas*.

The relationship between soil physicochemical parameters and fungal communities was studied for all groups. pH, organic matter and exchangeable potassium showed statistically significant differences. Similarly to the bacterial communities, CAP1 was the significant axis. CAP on Bray-Curtis dissimilarities was done and the two first axes explained 28.5% of the variance (Fig 6B). In this case, 41 fungal genera were strongly and significantly correlated to any of the analysed physicochemical parameters. The five genera shown in Fig 5 were affected by both pH and organic matter, except *Coniosporium* and *Solicoccozyma*, the former being positively correlated only with pH and the latter negatively correlated just with organic matter (S3 Table in S1 File).

## Discussion

Understanding the mechanisms shaping microbial communities in the rhizosphere of trees could improve the management of bioinoculants used in forestry and woody crops, as well as in plant fitness through biocontrol strategies and enhanced nutrient uptake. The aim of this study was to determine how, under the same pedological and climate conditions, rhizosphere and soil microbial communities are affected by tree species and soil management. Although this objective has been previously addressed in herbaceous species [40,43,74], our knowledge in the case of woody, long-living plants is scarce. Concerning the α-diversity indices in bacterial communities, non-significant value trend was observed when comparing soil management practices (i.e. no management *vs*. agronomical use), even though significant differences were

found between Holm oak rhizosphere soil and the rest of soils. That is, no rhizosphere effect was observed in olive roots. This result disagrees with most of the related literature [35,49,51,52], but agrees with a recent study performed at European scale that showed that the soil structure could be more important than soil management [44]. Particularly, only diversity indices, but not richness, showed differences in the above-mentioned case. This higher diversity in HOR could be due to the rhizosphere effect well documented for grass and herbaceous plants [74,75] but largely unknown for woody plants [45]. However, there was no rhizosphere effect in the agricultural soil (olive orchard) as indicated by α- and β-diversity analyses, likely due to the common soil management practices (i.e. soil plow after application of herbicides, pesticides and chemical fertilization) implemented in the orchard under study. Likewise, there was no difference between the rhizosphere of 'Frantoio' and 'Picual' trees. Thus, the effect of agrochemicals and management over bacterial communities cannot be underestimated.

In contrast, fungal communities showed an opposite pattern, HOR being the less diverse spot. It must be emphasized that our knowledge on the structure and functioning of rhizosphere fungi is still very limited [74]. Consequently, it is difficult to determine how fungal communities in this niche are shaped. It could be speculated that in non-agricultural soils and in the absence of soil structure disruption processes (e.g. no tillage), fungi could develop more stable mycelia networks enabling them to colonize and endure in the soil thereby preventing the establishment of other fungi. Regarding to this, it is worth mentioning that both HOB and HOR showed low fungal diversity. Thus, and as overall conclusion, our results show that fungal and bacterial communities responded differentially depending on the soil management. Indeed, bacterial diversity was higher in Holm oak rhizosphere and three groups (olive, Holm oak bulk soil and rhizosphere of Holm oak) could be detected. In contrast, fungal diversity was lower and only two groups of samples (olive and Holm oak) were differentiated. While how individual agronomical practices (i.e. irrigation, tillage, use of specific herbicides or pesticides, etc.), a combination of some of them, or all as a whole contribute to explain the observed differences will require further research efforts exceeding the objectives of this study. Moreover, when comparing Holm oak rhizosphere with olive rhizosphere soils, tree species represents a source of variation in the microbial community that should be considered in the design of upcoming studies. Predictably, bacterial diversity indices were higher than those for fungal communities, (four-fold in the case of the InvSimpson index). Our results agree with those from Urbanová *et al*. [76] who suggested that bacterial and fungal communities are not determined by the same environmental drivers. Moreover, these authors also showed that the effect of trees on the composition of the microbial community was stronger than soil properties, particularly in the case of fungi.

As opposed to the situation observed in the adjacent natural Holm oak neighborhood, soil management (i.e. the implementation of the usual agronomical practices in the region) was the main factor shaping microbial communities in the olive orchard. The prolonged implementation of these practices along time have likely also produced the decrease in organic matter content and the increase in pH in the olive orchard, compared to the situation found in the natural forestry soil. These two altered physicochemical parameters, together with exchangeable potassium in the particular case of fungal communities, were indeed responsible for the microbial shifts observed for most taxa when comparing forestry and agricultural soils. There is a general consensus in the literature [19,42,77,78] on the key role that pH plays to shape the structure of soil microbial communities. Our results show that the bacterial genera *Gemmatimonas* and *Candidate division WPS-1*, and also the fungal genera *Fusarium*, *Subramaniula* and *Stachybotris*, were positively correlated with pH, supporting their higher relative abundance in agricultural soils. According to this, *Gemmatimonas* was negatively affected by the organic matter content, a situation also observed for the fungi *Solicoccozyma*, *Fusarium*,

*Subramaniula* and *Stachybotris*. Nevertheless, other interesting genera were not affected by the soil physicochemical parameters evaluated in our study. This is the case of *Solirubrobacter*, *Rubrobacter*, *Gp7* and *Gp4*. *Solirubrobacter* was more abundant on Holm oak soil in comparison with olive soils, particularly in the rhizosphere. Little is known about the relationship of this genus with soil status. Recently, Sánchez-Marañón *et al.* [79] found that *Solirubrobacter* increased in more evolved Mediterranean soils, with very low content on carbonates. This agrees with our results, as Holm oak soil has a much lower level of carbonates than the soil present in the olive orchard. However, in contrast to our results, other studies reported higher abundance of this genus when an organic fertilizer (product of fermentation of filter mud, plant residues and molasses) was added to banana crops [80,81]. In contrast, abundance of *Solirubrobacter* diminished when the organic fertilizer was in excess (80 Tm per year and hectare of thermophilic digested sewage sludge) in carbonate-rich soils with cultivate oats [82]. In our case, this genus could be affected by an excess of inorganic fertilization, a situation often found in many olive orchards in the region subjected to traditional management practices, and that is obviously absent in the neighboring Holm oak soil. Moreover, the literature supports that *Rubrobacter* is found in extreme environments, such as contaminated or arid soils (especially with zinc) [40,83–85]. We observed that *Rubrobacter* was more represented in agricultural soil (OLR *vs*. HOR), what may be explained by the use of pesticides and other agrochemicals used in the olive orchard.

Rhizosphere effect was only observed in bacterial communities from natural conditions; that is, in the Holm oak rhizosphere. Despite being a clearly studied effect on both native and agricultural soils [19,74,78,86], our data do not support such an effect in the olive grove under study. A possible explanation would be the intense agronomical management to which the orchard soil is subjected (see S1 Fig in S1 File). Regarding this effect in native soils, *Solirubrobacter* was the only genus more abundantly represented in the Holm oak rhizosphere, being *Gp6* equally represented in olive samples and solely depleted in Holm oak bulk soils.

Plant genotype is a key factor in shaping belowground microbial communities [87,88], and olive trees are not the exception [39]. Remarkably, differences were found neither in bacterial nor in fungal rhizosphere communities of 'Frantoio' and 'Picual' trees under conditions here evaluated, which includes substitution of 'Picual' trees affected by verticilosis. It is tempting to speculate that, under intensive soil management practices implemented in the orchard (i.e. excessive use of agrochemicals, tillage, etc.), the expected cultivar/genotype-driven differences of rhizosphere microbial communities are blurred. It remained to be evaluated whether the same situation could be found in the olive root endosphere since is a more selective compartment for microorganisms because the interactions with the plant are narrower than in the rhizosphere [89]. Minimal differences in the rhizosphere microbial communities have also been reported in other tree species (beech and Norway spruce) grown in native soils [45]. In our study, the rhizosphere of Holm oak and olive trees shared 99 bacterial and 13 fungal genera. Since it is known that bacterial diversity is higher than fungal diversity, our result was somehow expected. However, while the core bacteriome was composed by more than 50% of the total sequences, the core mycobiome only have about 20% of the sequences. Although a common pattern was observed with very few specific genera for each tree species and with a low relative abundance. Thus, the main differences between the treatments (HOR, HOB, OLB, Pic and Fra) were due to the differences on the relative abundance of each genus in each sample. This situation is similar to that reported for microbial communities in the comparison between Norway spruce and beech [45]. It seems that most of the native microbial community was selected from the original soil after agricultural management, but driving the loss of rhizosphere effect which remained present only in the native (forestry) soil.

## Conclusions

In summary, our results show that: (i) the effect of the plant was only detected in the Holm oak rhizosphere; (ii) bulk olive and native bulk soils showed differences in their microbial community compositions; and (iii) only minor differences were found between bulk and olive rhizosphere soils. Therefore, the hypothesis that the olive-associated soil microbiota is shaped depending on the VWO susceptibility level of the olive cultivars could not be proven, at least in the orchard here under study. Thus, tolerance of olive cultivars to *V. dahliae* under the specific field and agronomical conditions analyzed must rely on other major factors [14,90] than the structure and composition of the rhizosphere microbial communities. However, since root endophytic microbial communities in trees are likely more protected from the effects of external factors (e.g. fluctuating environmental conditions, soil management practices, etc.) than those residing in the rhizosphere/bulk soils, their study in relation with VWO differential tolerance of olive cultivars will deserve attention of our future studies.

## Supporting information

**S1 File.**
(PDF)

## Acknowledgments

We thank Antonio Estévez (Nutesca, S.L.) for guidance to identify olive orchards and assistance to identify olive cultivars during sampling.

## Author Contributions

**Conceptualization:** Jesús Mercado-Blanco, Manuel Fernández-López.

**Formal analysis:** Antonio J. Fernández-González, Nuria M. Wentzien, Pablo J. Villadas, Ana V. Lasa.

**Funding acquisition:** Jesús Mercado-Blanco, Manuel Fernández-López.

**Investigation:** Antonio J. Fernández-González, Pablo J. Villadas, Antonio Valverde-Corredor, Carmen Gómez-Lama Cabanás.

**Project administration:** Jesús Mercado-Blanco, Manuel Fernández-López.

**Supervision:** Manuel Fernández-López.

**Validation:** Antonio J. Fernández-González, Nuria M. Wentzien, Manuel Fernández-López.

**Writing – original draft:** Antonio J. Fernández-González, Nuria M. Wentzien, Jesús Mercado-Blanco, Manuel Fernández-López.

**Writing – review & editing:** Manuel Fernández-López.

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
