## [Decision Letter · Decision Letter 0]

16 Jun 2020

PONE-D-20-12873

Comparative study of neighboring Holm-oak and olive trees- belowground microbial communities subjected to different soil management

PLOS ONE

Dear Dr. Fernández-López,

Thank you for submitting your manuscript to PLOS ONE. After careful consideration, we feel that it has merit but does not fully meet PLOS ONE’s publication criteria as it currently stands. Therefore, we invite you to submit a revised version of the manuscript that addresses the points raised during the review process.

Authors are invited to carefully follow suggestions from the Reviewer in order to improve the ms and render it acceptable for publicatin.

We look forward to receiving your revised manuscript.

Kind regards,

Sabrina Sarrocco

Academic Editor

PLOS ONE

Journal Requirements:

3. We note that you are reporting an analysis of a microarray, next-generation sequencing, or deep sequencing data set. PLOS requires that authors comply with field-specific standards for preparation, recording, and deposition of data in repositories appropriate to their field. Please upload these data to a stable, public repository (such as ArrayExpress, Gene Expression Omnibus (GEO), DNA Data Bank of Japan (DDBJ), NCBI GenBank, NCBI Sequence Read Archive, or EMBL Nucleotide Sequence Database (ENA)). In your revised cover letter, please provide the relevant accession numbers that may be used to access these data. For a full list of recommended repositories, see http://journals.plos.org/plosone/s/data-availability#loc-omics or http://journals.plos.org/plosone/s/data-availability#loc-sequencing.

5. We note that Figures S1Fig and S2Fig in your submission contain satellite images which may be copyrighted. All PLOS content is published under the Creative Commons Attribution License (CC BY 4.0), which means that the manuscript, images, and Supporting Information files will be freely available online, and any third party is permitted to access, download, copy, distribute, and use these materials in any way, even commercially, with proper attribution. For these reasons, we cannot publish previously copyrighted maps or satellite images created using proprietary data, such as Google software (Google Maps, Street View, and Earth). For more information, see our copyright guidelines: http://journals.plos.org/plosone/s/licenses-and-copyright.

5.1. You may seek permission from the original copyright holder of Figures S1Fig and S2Fig to publish the content specifically under the CC BY 4.0 license.

5.2. If you are unable to obtain permission from the original copyright holder to publish these figures under the CC BY 4.0 license or if the copyright holder’s requirements are incompatible with the CC BY 4.0 license, please either i) remove the figure or ii) supply a replacement figure that complies with the CC BY 4.0 license. Please check copyright information on all replacement figures and update the figure caption with source information. If applicable, please specify in the figure caption text when a figure is similar but not identical to the original image and is therefore for illustrative purposes only.

6. Please upload a copy of Figure 6, to which you refer in your text on page 2. If the figure is no longer to be included as part of the submission please remove all reference to it within the text.

Reviewers' comments:

Reviewer's Responses to Questions

**Comments to the Author**

1. Is the manuscript technically sound, and do the data support the conclusions?

Reviewer #1: Yes

2. Has the statistical analysis been performed appropriately and rigorously? 

Reviewer #1: Yes

3. Have the authors made all data underlying the findings in their manuscript fully available?

Reviewer #1: Yes

4. Is the manuscript presented in an intelligible fashion and written in standard English?

Reviewer #1: Yes

5. Review Comments to the Author

Reviewer #1: This report describes bacterial and fungal communities associated with bulk soil and with tree roots at a single location. Replicates represent separately sampled trees. The work appears to be well done, and useful information is reported.

One weakness of the site & experimental design is a confounding of plant identity with management history/soil properties when Holm oak rhizosphere samples are contrasted with olive rhizosphere samples. I think this confounding should be highlighted more frequently and explicitly in the discussion. Are there also topographical or other aspects of microsite that are confounded in this contrast? (For instance, was there some soil or topographical feature that influenced where the planting of olive trees ceased?) Sites within metres of each other may be ‘worlds apart’ microbiologically speaking, if the soil properties are different.

I appreciate the effort that was made to ensure that the collected soil samples were representative.

Given how the study is set up and motivated, some mention should be given to the detection of Verticillium in the sequence data (including if there was no detection of Verticillium!).

Please clarify how the data were treated to create the ‘OL’ aggregation. Was this treated as n = 10 (5 Fra samples + 5 Pic samples)? How does this relate to the description in the S4 Fig. legend: “OL: average of Frantoio and Picual rhizospheres”?

Figure 6 is missing from the submission!

- I would encourage the authors to adjust the wording of the title so that the use of dashes is not required. Throughout, I don’t think there needs to be a dash in ‘Holm-oak’.

- There is a spelling error in the short title

- Ln 27 (and elsewhere): the term ‘next generation sequencing’ does not age well. The ‘next generation sequencing’ of 10 years ago is already obsolete…

- Ln 27: I think the acronyms would be easier for the reader if you used HOR for ‘Holm oak rhizosphere’ and OLR ‘olive tree rhizosphere’

- Ln 169: there is no utility in defining an acronym that never gets used again in the paper

- Ln 188: S2 Fig should not be referenced before S1 Fig.

- Ln 189: ‘massive sequencing’ should be ‘massively parallel sequencing’

- Ln 217: Please indicate concisely in the results how the observed community profile for the reference community matched the expected profile. Were negative control samples also included? (i.e., blank DNA extractions, carried all the way through sequencing)

- Ln 267, 470 & elsewhere: the term ‘bacteriome’ here is an unusual usage; generally, this term refers to a specialized organ within insects that houses microbial symbionts.

- Ln 267 (and throughout): ‘core taxa’ might be preferable to ‘cores’

- Ln 282 and throughout: the Bray-Curtis index is correctly referred to as a dissimilarity (i.e., bounded at 1 as a maximum), but the UniFrac scores are distances (also at Ln 347 and elsewhere). Methods should be described for building the tree from which UniFrac scores were calculated.

- Ln 543: are these tree roots woody? There may only be active exudation at the root tips or specific regions within the root system.

Fig. 1: The y-axis labels should appear next to the y-axis, rather than as a title above each panel. The legend should indicate which summary statistics are indicated by the boxplots (e.g., is the centre line the mean or the median?). Either each individual data point should be made visible, or the legend should indicate the sample size. Statistically significant differences, if any, should be indicated on the figure. What are the error bars around the individual outlying points in the A; Chao1 panel? The order of the treatments could be improved; for instance, currently rhizosphere samples are before bulk soil samples for Holm oak, but the order is opposite for the olive tree samples.

Fig. 2: The legend incorrectly refers to these as ‘principal components’ ordinations (they are principal coordinates ordinations). Colours appear to differ slightly between panels for “OLB” and “Pic” treatments.

Fig. 3: The order of the treatments could be improved; for instance, currently rhizosphere samples are before bulk soil samples for Holm oak, but the order is opposite for the olive tree samples. I thought the sequences originating from chloroplasts were culled, yet chloroplast still shows up here?

Fig. 4, Fig. 5: ‘Statistically significant genera’ is not a meaningful phrase; it is actually ‘genera whose relative abundance differed significantly’. The figure legends should indicate the statistical test. Genus names should be italicized.

Supplementary Material:

In the table of soil properties, the precision with which values are reported should be consistent across samples.

6. PLOS authors have the option to publish the peer review history of their article (what does this mean?). If published, this will include your full peer review and any attached files.

Reviewer #1: No

---

## [Author Response · Author response to Decision Letter 0]

10 Jul 2020

First at all I would like to grateful the work of reviewer and editor on this manuscript which will help us to improve it. Please find below our answer to your comments point by point. The line numbers are those of the marked-up copy of the manuscript.

Editorial comments.

Thanks for the advice. We have checked again the manuscript and the file names to meet the style requirements.

The sampling site was part of a private farm, in order to point out this situation at lines 171-174 we had included the sentences: “Permission to get access to the orchard was granted by the owner of the farm, who was informed of the sampling activities in advance. Additional permits from local, regional or national authorities were not needed since no attempt to collect genetic resources was intended.“. 

3. We note that you are reporting an analysis of a microarray, next-generation sequencing, or deep sequencing data set. PLOS requires that authors comply with field-specific standards for preparation, recording, and deposition of data in repositories appropriate to their field. Please upload these data to a stable, public repository (such as ArrayExpress, Gene Expression Omnibus (GEO), DNA Data Bank of Japan (DDBJ), NCBI GenBank, NCBI Sequence Read Archive, or EMBL Nucleotide Sequence Database (ENA)).

Our data from next-generation sequencing are already deposited in the NCBI Sequence Read Archive. In the manuscript, at the end of the Material and method section, you can find: 

Data publicly available

The datasets generated and analyzed during the current study are available in the NCBI Sequence Read Archive (SRA) under the BioProject number PRJNA510560.

4. We note that you have included the phrase “data not shown” in your manuscript. Unfortunately, this does not meet our data sharing requirements. PLOS does not permit references to inaccessible data. We require that authors provide all relevant data within the paper, Supporting Information files, or in an acceptable, public repository.

According to this comment, we have included the p-values of the analyses and now the sentence is corrected to: “In agreement with β-diversity, post hoc test showed no significant differences (Benjamini-Hochberg FDR corrected p-values > 0.05) among phyla and genera when comparing Pic and Fra rhizosphere soils.” in lines 435-437.

5. We note that Figures S1Fig and S2Fig in your submission contain satellite images which may be copyrighted. All PLOS content is published under the Creative Commons Attribution License (CC BY 4.0), which means that the manuscript, images, and Supporting Information files will be freely available online, and any third party is permitted to access, download, copy, distribute, and use these materials in any way, even commercially, with proper attribution. For these reasons, we cannot publish previously copyrighted maps or satellite images created using proprietary data, such as Google software (Google Maps, Street View, and Earth). For more information, see our copyright guidelines: http://journals.plos.org/plosone/s/licenses-and-copyright.

Thank you very much for the advice and the links to other maps sources; it will be very useful for our future works. In Figures S1Fig and S2Fig, we have change the satellite images obtained from Google Earth by others from USGS EROS (Earth Resources Observatory and Science (EROS) Center): http://eros.usgs.gov/# , specifically at webpage https://earthexplorer.usgs.gov/ . The origin and link of the images are point out in the figures legend.

5. Review Comments to the Author

Reviewer #1: This report describes bacterial and fungal communities associated with bulk soil and with tree roots at a single location. Replicates represent separately sampled trees. The work appears to be well done, and useful information is reported.

One weakness of the site & experimental design is a confounding of plant identity with management history/soil properties when Holm oak rhizosphere samples are contrasted with olive rhizosphere samples. I think this confounding should be highlighted more frequently and explicitly in the discussion. Are there also topographical or other aspects of microsite that are confounded in this contrast? (For instance, was there some soil or topographical feature that influenced where the planting of olive trees ceased?) Sites within metres of each other may be ‘worlds apart’ microbiologically speaking, if the soil properties are different.

In lines 605-608 we have added the sentence: “Moreover, when comparing Holm oak rhizosphere with olive rhizosphere soils, tree species represents a source of variation in the microbial community that should be considered in the design of upcoming studies”.

There are no other different aspects between the comparison sites apart from tree species and soil management. The natural field was similar to that managed but remained unaltered only for government regulation for protection of natural resources as wild trees.

I appreciate the effort that was made to ensure that the collected soil samples were representative.

Given how the study is set up and motivated, some mention should be given to the detection of Verticillium in the sequence data (including if there was no detection of Verticillium!).

We agree the reviewer’s comment and we have added at lines 492-494 the sentence: “Moreover, it has to be mentioned that Verticillium was no detected in spite of the deep sequencing analysis.”.

Please clarify how the data were treated to create the ‘OL’ aggregation. Was this treated as n = 10 (5 Fra samples + 5 Pic samples)? How does this relate to the description in the S4 Fig. legend: “OL: average of Frantoio and Picual rhizospheres”?

In order to avoid confusion when reading the paper, we have included a few lines to explain how this OL (now OLR) aggregate was treated. First, in line 440-441, in order to clarify the use of this procedure in the relative abundance calculation. Second, in line 275 to explain it in relation to construction of core microbiomes. Finally, S4 Fig and S3 Fig. legends have been properly changed as well. 

Figure 6 is missing from the submission!

Our apologies for the omission. It has been fixed.

I would encourage the authors to adjust the wording of the title so that the use of dashes is not required. Throughout, I don’t think there needs to be a dash in ‘Holm-oak’.

Considering the reviewer recommendation, “Holm-oak” has been substituted by “holm oak” throughout the whole text. 

There is a spelling error in the short title

Yes, it’s our mistake; it should be “holm oak” and not “holm aok”. Thanks for the advice. 

Ln 27 (and elsewhere): the term ‘next generation sequencing’ does not age well. The ‘next generation sequencing’ of 10 years ago is already obsolete…

We completely agree with the reviewer point of view: NGS has been changed to high-throughput sequencing. 

Ln 27: I think the acronyms would be easier for the reader if you used HOR for ‘Holm oak rhizosphere’ and OLR ‘olive tree rhizosphere’

We truly thank your recommendation, as we consider it to make much clearer our research paper. Changes have been applied to the text and figures. 

Ln 169: there is no utility in defining an acronym that never gets used again in the paper.

The acronym “m.a.s.l.” has been substituted by “meters above sea level”, now at line 170.

Ln 188: S2 Fig should not be referenced before S1 Fig.

We consider it may be a confusion. S1 Fig is referenced for the first time in line 171 while S2 Fig is referenced afterwards, in line 187.

Ln 189: ‘massive sequencing’ should be ‘massively parallel sequencing’

The recommendation has been taking into account and changed properly. 

Ln 217: Please indicate concisely in the results how the observed community profile for the reference community matched the expected profile. Were negative control samples also included? (i.e., blank DNA extractions, carried all the way through sequencing).

The information about the reference community has been included in a new paragraph in lines 342-350. Moreover, the sequencing service always includes negative controls and they only yielded a few hundreds of reads.

Ln 267, 470 & elsewhere: the term ‘bacteriome’ here is an unusual usage; generally, this term refers to a specialized organ within insects that houses microbial symbionts.

We would like to explain and clarify the meaning of ‘bacteriome’ because we understand it could lead to confusion. In our context (plant microbiomes), bacteriome and mycobiome are used to make a distinction between the cores microbiomes obtained from 16S rRNA sequences and ITS2 sequences, respectively. Although bacteriome could have another meaning, we consider that, in the context of our research, it should be understood as bacterial community. May we mention a bunch of articles that use this term likely as we do: doi:10.1016/j.coviro.2019.05.007; doi:10.1016/j.metabol.2017.04.014; doi: 10.1128/mBio.01250-16; doi: 10.1016/S0016-5085; doi: 10.3389/fmicb.2018.00077. Moreover, we have already published articles like this https://doi.org/10.1186/s40168-020-0787-2, where we use this term with the meaning we intended to. 

Ln 267 (and throughout): ‘core taxa’ might be preferable to ‘cores’

The changed proposed by the reviewer has been included in the paper.

Ln 282 and throughout: the Bray-Curtis index is correctly referred to as a dissimilarity (i.e., bounded at 1 as a maximum), but the UniFrac scores are distances (also at Ln 347 and elsewhere). Methods should be described for building the tree from which UniFrac scores were calculated.

The reviewer is right, we apologize for the mistake and thanks for the advice. We changed UniFrac dissimilarities by distances according to his/her recommendation. In addition, we have described the methods for obtaining the phylogenetic tree for UniFrac scores calculation in lines 292-294.

Ln 543: are these tree roots woody? There may only be active exudation at the root tips or specific regions within the root system.

We appreciate the comment, as it is a very interesting point to take into account, and that must be considered in the sampling process. In this sense, we mentioned this topic in the Methods section, specifically between lines 184-186: ‘soil samples were collected (5 to 20-cm depth) following the main roots of each plant until finding cork-free roots’. This procedure was done trying to obtain active roots, in the same way that in our previous research with woody plants. 

Fig. 1: The y-axis labels should appear next to the y-axis, rather than as a title above each panel. The legend should indicate which summary statistics are indicated by the boxplots (e.g., is the centre line the mean or the median?). Either each individual data point should be made visible, or the legend should indicate the sample size. Statistically significant differences, if any, should be indicated on the figure. What are the error bars around the individual outlying points in the A; Chao1 panel? The order of the treatments could be improved; for instance, currently rhizosphere samples are before bulk soil samples for Holm oak, but the order is opposite for the olive tree samples.

We would like to answer with a list to make the explanation clearer:

• The y-axis labels have been changed in order to appear next to the axis.

• Fig. 1’s legend has been rewritten to include the summary statistics and the sample size. 

• Statistically significant differences have been indicated on the figure. 

• We perfectly understand the reviewer’s suggestion regarding the treatments order, but we would like to point out the reasons behind this specific structure. While analyzing the data and producing the figures, we noticed that the differences between the bulk soils were more subtle than the ones observed among rhizospheres soils. Considering this, we opted to rearrange the figures and display bulk soils together, thinking that this could make the interpretation easier to the reader. 

Fig. 2: The legend incorrectly refers to these as ‘principal components’ ordinations (they are principal coordinates ordinations). Colours appear to differ slightly between panels for “OLB” and “Pic” treatments.

As the reviewer correctly points out, principal components has been substituted by principal coordinates. Fig.2 colours have been fixed as well and small modifications were made to improve the quality. 

Fig. 3: The order of the treatments could be improved; for instance, currently rhizosphere samples are before bulk soil samples for Holm oak, but the order is opposite for the olive tree samples. I thought the sequences originating from chloroplasts were culled, yet chloroplast still shows up here?

Our reasons for maintaining this order of treatments has already been explained above in Fig. 1 comments and also to provide the same order along the article. 

Regarding chloroplasts appearance, there are sequences of some genera that can be classified as Cyanobacteria/Chloroplast at phylum level, e.g GpXIII, GpI or Cyanobacteria_unclassified. All chloroplasts appearances in the sequence data were individually analyzed and those where Cyanobacteria/Chloroplast were classified as Streptophyta later on were deleted. This is the reason why we can find Cyanobacteria/Chloroplast at phylum level, not being a chloroplast sequence at all. 

Fig. 4, Fig. 5: ‘Statistically significant genera’ is not a meaningful phrase; it is actually ‘genera whose relative abundance differed significantly’. The figure legends should indicate the statistical test. Genus names should be italicized.

Following the reviewer’s recommendation, Fig.4 and Fig.5 legends have been changed to include the statistical test and genera names have been italicized.

Supplementary Material:

In the table of soil properties, the precision with which values are reported should be consistent across samples.

With the changes applied considering the reviewer’s suggestion, the precision across samples should be consistent. Moreover, the picture of S1 and S2 Figures is changed and the map’s source is provided in the legend.

---

## [Editor Report · Decision Letter 1]

15 Jul 2020

Comparative study of neighboring Holm oak and olive trees-belowground microbial communities subjected to different soil management

PONE-D-20-12873R1

Dear Dr. Fernández-López,

We’re pleased to inform you that your manuscript has been judged scientifically suitable for publication and will be formally accepted for publication once it meets all outstanding technical requirements.

Kind regards,

Sabrina Sarrocco

Academic Editor

PLOS ONE

Additional Editor Comments (optional):

Authors modified the text according to Reviwer's suggestions. the paper is now accepted for publication in PlosOne.
---

## [Editor Report · Acceptance letter]

30 Jul 2020

PONE-D-20-12873R1 

Comparative study of neighboring Holm oak and olive trees- belowground microbial communities subjected to different soil management 

Dear Dr. Fernández-López:

I'm pleased to inform you that your manuscript has been deemed suitable for publication in PLOS ONE. Congratulations! Your manuscript is now with our production department. 

Kind regards, 

on behalf of

Dr Sabrina Sarrocco 

Academic Editor

PLOS ONE